# LEARNING MIRROR MAPS IN POLICY MIRROR DESCENT

**Carlo Alfano**[*]
Department of Statistics
University of Oxford

**Sebastian Towers**[†]
FLAIR
University of Oxford

**Silvia Sapora**[†]
FLAIR, Department of Statistics
University of Oxford

**Chris Lu**
FLAIR
University of Oxford

**Patrick Rebeschini**
Department of Statistics
University of Oxford

## ABSTRACT

Policy Mirror Descent (PMD) is a popular framework in reinforcement learning, serving as a unifying perspective that encompasses numerous algorithms. These algorithms are derived through the selection of a mirror map and enjoy finite-time convergence guarantees. Despite its popularity, the exploration of PMD's full potential is limited, with the majority of research focusing on a particular mirror map—namely, the negative entropy—which gives rise to the renowned Natural Policy Gradient (NPG) method. It remains uncertain from existing theoretical studies whether the choice of mirror map significantly influences PMD's efficacy. In our work, we conduct empirical investigations to show that the conventional mirror map choice (NPG) often yields less-than-optimal outcomes across several standard benchmark environments. Using evolutionary strategies, we identify more efficient mirror maps that enhance the performance of PMD. We first focus on a tabular environment, i.e. Grid-World, where we relate existing theoretical bounds with the performance of PMD for a few standard mirror maps and the learned one. We then show that it is possible to learn a mirror map that outperforms the negative entropy in more complex environments, such as the MinAtar suite. Additionally, we demonstrate that the learned mirror maps generalize effectively to different tasks by testing each map across various other environments.

## 1 INTRODUCTION

Policy gradient (PG) methods (Williams & Peng, 1991; Sutton et al., 1999; Konda & Tsitsiklis, 2000; Baxter & Bartlett, 2001) are some of the most widely-used mechanisms for policy optimization in reinforcement learning (RL). These algorithms are gradient-based methods that optimize over a class of parameterized policies and have become a popular choice for RL problems, both in theory (Kakade, 2002; Peters & Schaal, 2008; Bhatnagar et al., 2009; Schulman et al., 2015; Mnih et al., 2016; Schulman et al., 2017; Lan, 2022a) and in practice (Shalev-Shwartz et al., 2016; Berner et al., 2019; Ouyang et al., 2022).

Among PG methods, some of the most successful algorithms are those that employ some form of regularization in their updates, ensuring that the newly updated policy retains some degree of similarity to its predecessor. This principle has been implemented in different ways. For instance, trust region policy optimization (TRPO) Schulman et al. (2015) imposes a Kullback-Leibler divergence Kullback & Leibler (1951) hard constraint for its updates, while proximal policy optimization (PPO) (Schulman et al., 2017) uses a clipped objective to penalize large updates. A framework that has recently attracted attention and belongs to this heuristic is that of policy mirror descent (PMD) (Tomar et al., 2022; Lan, 2022a; Xiao, 2022; Kuba et al., 2022; Vaswani et al., 2022; Alfano et al., 2023), which applies mirror descent Nemirovski & Yudin (1983) to RL to regularize the policy updates.

PMD consists of a wide class of algorithms, each derived by selecting a *mirror map* that introduces distinct regularization characteristics. In recent years, PMD has been investigated through numerical

---

[*]Correspondence to carlo.alfano@stats.ox.ac.uk.
[†]Equal contribution.

experiments (Tomar et al., 2022), but it has mainly been analysed from a theoretical perspective. To the best of our knowledge, research has been mostly focused either on the particular case of the negative entropy mirror map, which generates the natural policy gradient (NPG) algorithm (Kakade, 2002; Agarwal et al., 2021), or on finding theoretical guarantees for a generic mirror map.

NPG has been proven to converge to the optimal policy, up to a difference in expected return (or *error floor*), in several settings, e.g. using tabular, linear, general parameterization (Agarwal et al., 2021) or regularizing rewards (Cen et al., 2021). It has been shown that NPG benefits from implicit regularization (Hu et al., 2022), that it can exploit optimism (Zanette et al., 2021; Liu et al., 2023), and some of its variants have been evaluated in simulations (Vaswani et al., 2022; 2023). When considering other specific mirror maps investigated in the RL literature, the Tsallis entropy has been noted for enhancing performance in offline settings (Tomar et al., 2022) and for offering improved sample efficiency in online settings (Li & Lan, 2023), when compared to the negative entropy.

There is a substantial body of theoretical research focused on the general case of mirror maps. This research demonstrates that PMD achieves convergence to the optimal policy under the same conditions as NPG, as evidenced by various studies (Xiao, 2022; Lan, 2022b; Yuan et al., 2023; Alfano et al., 2023). Except for the setting where we have access to the true value of the policy, convergence guarantees in stochastic settings are subject to an error floor due to the inherent randomness or bias within the algorithm. In the majority of PMD analyses involving generic mirror maps, both the convergence rate and the error floor show mild dependence on the specific choice of mirror map. Typically, the effect of the mirror map appears explicitly as a multiplicative factor in the convergence rate and it appears implicitly in the error floor. These analyses often rely on *upper bounds*, meaning they may not accurately reflect the algorithms' actual performance in applications.

In this work, we contribute to the literature with an empirical investigation of PMD, with the objectives of finding a mirror map that consistently outperforms the negative entropy mirror map and of understanding how the theoretical guarantees of PMD relate to simulations. We first consider a set of tabular environments, i.e. Grid-World (Oh et al., 2020), where we compare the empirical results to the theoretical guarantees given by Xiao (2022), which we can compute as we have full control over the environment. In this setting, we provide a learned mirror map which outperforms the negative entropy in all tested environments. Our experiments suggest that the error floor appearing in prototypical PMD convergence guarantees is not a good performance indicator: our learned mirror map achieves the best value while presenting the worst theoretical error floor, implying that the *upper* bounds typically considered in the PMD literature are loose with respect to the choice of the mirror map. Additionally, our experiments indicate that having small policy updates leads to smoother value improvements over time with less instances of performance degradation, as suggested by the monotonic policy improvement property given by Xiao (2022). We then consider two non-tabular settings, i.e. the Basic Control Suite and the MinAtar Suite, which are more realistic but also more complex, and therefore prevent us from computing the exact theoretical guarantees. We learn a mirror map for each of these environments and, also in this case, show that the learned mirror maps lead to a higher performance of PMD than the negative entropy. Moreover, we show that the learned mirror maps generalize well to other tasks, by testing each of the learned mirror maps on all the other environments. Lastly, we tackle continuous control tasks in MuJoCo (Todorov et al., 2012), where we show that the mirror map learned on one environment surpasses the negative entropy across several environments.

To establish our findings, we employ the standard formulation of PMD (Xiao, 2022) for the tabular setting and the continuous control tasks, and we used a generalized version, Approximate Mirror Policy Optimization (AMPO) (Alfano et al., 2023), for the non-tabular setting with discrete action spaces. To allow optimization over the space of mirror maps, we introduce parameterization schemes for mirror maps, one for PMD and one for AMPO. Specifically, we propose a parameterization for $\omega$-potentials, which have been shown to induce a wide class of mirror maps (Krichene et al., 2015). We use evolution strategies (ES) to search for the mirror map that maximizes the performance of PMD and AMPO over an environment.

AMPO (Alfano et al., 2023) is a recently-proposed PMD framework designed to integrate general parameterization schemes, in our case neural networks, and arbitrary mirror maps. It benefits from theoretical guarantees, as Alfano et al. (2023) show that AMPO has quasi-monotonic updates as well as sub-linear and linear convergence rates, depending on the step-size schedule. These desirable properties make AMPO particularly suitable for our numerical investigation.

ES are a type of population-based stochastic optimization algorithm that leverages random noise to generate a diverse pool of candidate solutions, and have been successfully applied to a variety of tasks (Real et al., 2019; Salimans et al., 2017; Such et al., 2018). The main idea consists in iteratively selecting higher-performing individuals, w.r.t. a fitness function, resulting in a gradual convergence towards the optimal solution. ES algorithms are gradient-free and have been shown to be well-suited for optimisation problems where the objective function is noisy or non-differentiable and the search space is large or complex (Beyer, 2000; Lu et al., 2022; 2023). We use ES to search over the parameterized classes of mirror maps we introduce, by defining the fitness of a particular mirror map as the value of the last policy outputted by PMD and AMPO, for fixed hyper-parameters.

The rest of the paper is organized as follows. In Section 2, we introduce the setting of RL as well as the PMD and AMPO algorithms. We describe the methodology behind our numerical experiments in Section 3, which are then discussed in Section 4. Finally, we discuss related works from the literature on automatic discovery of machine learning algorithms in Appendix A and give our conclusions in Section 5.

## 2 PRELIMINARIES

### 2.1 REINFORCEMENT LEARNING

Define a discounted Markov Decision Process (MDP) as the tuple $\mathcal{M} = (\mathcal{S}, \mathcal{A}, P, r, \gamma, \mu)$, where $\mathcal{S}$ and $\mathcal{A}$ are respectively the state and action spaces, $P(s' \mid s, a)$ is the transition probability from state $s$ to $s'$ when taking action $a$, $r(s, a) \in [0, 1]$ is the reward function, $\gamma$ is a discount factor, and $\mu$ is a starting state distribution. A *policy* $\pi \in (\Delta(\mathcal{A}))^{\mathcal{S}}$, where $\Delta(\mathcal{A})$ is the probability simplex over $\mathcal{A}$, represents the behavior of an agent on an MDP, whereby at state $s \in \mathcal{S}$ the agents takes actions according to the probability distribution $\pi(\cdot \mid s)$.

Our objective is for the agent to find a policy that maximizes the expected discounted cumulative reward for the starting state distribution $\mu$. That is, we want to find

$$\pi^{\star} \in \underset{\pi \in (\Delta(\mathcal{A}))^{\mathcal{S}}}{\operatorname{argmax}} \mathbb{E}_{s \sim \mu}[V^{\pi}(s)]. \tag{1}$$

Here $V^{\pi} : \mathcal{S} \to \mathbb{R}$ denotes the *value function* associated with policy $\pi$ and is defined as

$$V^{\pi}(s) := \mathbb{E}\Big[ \sum\nolimits_{t=0}^{\infty} \gamma^t r(s_t, a_t) \mid \pi, s_0 = s \Big],$$

where $s_t$ and $a_t$ are the current state and action at time $t$ and the expectation is taken over the trajectories generated by $a_t \sim \pi(\cdot|s_t)$ and $s_{t+1} \sim P(\cdot|s_t, a_t)$.

Similarly to the value function, we define the $Q$-function associated with a policy $\pi$ as

$$Q^{\pi}(s, a) := \mathbb{E}\Big[ \sum\nolimits_{t=0}^{\infty} \gamma^t r(s_t, a_t) \mid \pi, s_0 = s, a_0 = a \Big],$$

where the expectation is once again taken over the trajectories generated by the policy $\pi$. When the state and action spaces are finite, the $Q$-function can be expressed as $Q^{\pi} = (I - \gamma P^{\pi})^{-1} r$, where $P^{\pi}$ is a square matrix where the position $((s, a), (s', a'))$ is occupied by $\pi(a'|s')P(s'|s, a)$. We also define the discounted state visitation distribution as

$$d_{\mu}^{\pi}(s) := (1 - \gamma)\mathbb{E}_{s_0 \sim \mu}\Big[ \sum\nolimits_{t=0}^{\infty} \gamma^t P(s_t = s \mid \pi, s_0) \Big],$$

where $P(s_t = s \mid \pi, s_0)$ represents the probability of the agent being in state $s$ at time $t$ when following policy $\pi$ and starting from $s_0$. The probability distribution over states $d_{\mu}^{\pi}(s)$ represents the proportion of time spent on state $s$ when following policy $\pi$.

### 2.2 POLICY MIRROR DESCENT

We review the PMD framework, starting from mirror maps (Bubeck, 2015). Let $\mathcal{X} \subseteq \mathbb{R}^{\mathcal{A}}$ be a convex set. A *mirror map* $h : \mathcal{X} \to \mathbb{R}$ is a strictly convex, continuously differentiable and essentially smooth function[1] that satisfies $\nabla h(\mathcal{X}) = \mathbb{R}^{\mathcal{A}}$. In particular, we consider mirror maps belonging to the $\omega$-potential mirror map class, which contains most mirror maps used in the literature.

---

[1]A function $h$ is *essentially smooth* if $\lim_{x \to \partial \mathcal{X}} \|\nabla h(x)\|_2 = +\infty$, where $\partial \mathcal{X}$ denotes the boundary of $\mathcal{X}$.

**Definition 2.1** (ω-*potential mirror map* ([Krichene et al., 2015](#))). *For* $u \in (-\infty, +\infty]$, $\omega \leq 0$, *an* ω-*potential is defined as an increasing* $C^1$-*diffeomorphism* $\phi : (-\infty, u) \to (\omega, +\infty)$ *such that*

$$\lim_{x \to -\infty} \phi(x) = \omega, \ \lim_{x \to u} \phi(x) = +\infty, \ \int_0^1 \phi^{-1}(x)dx \leq \infty.$$

For any ω-potential φ, we define the associated mirror map $h_\phi$ as

$$h_\phi(\pi_s) = \sum_{a \in \mathcal{A}} \int_1^{\pi(a|s)} \phi^{-1}(x)dx.$$

When $\phi(x) = e^x$ we recover the negative entropy mirror map, which is the standard choice of mirror map in the RL literature ([Agarwal et al., 2021](#); [Tomar et al., 2022](#); [Hu et al., 2022](#); [Vaswani et al., 2022](#); [2023](#)), while we recover the $\ell_2$-norm when $\phi(x) = x$ (see Appendix [B.1](#)). Mirror maps belonging to this class are particularly advantageous as they can be defined by a single scalar function $\phi$, without having to account for the dimension of the action space $\mathcal{A}$. The *Bregman divergence* ([Bregman, 1967](#); [Censor & Zenios, 1997](#)) induced by the mirror map $h$ is defined as

$$\mathcal{D}_h(x, y) := h(x) - h(y) - \langle \nabla h(y), x - y \rangle,$$

where $\mathcal{D}_h(x, y) \geq 0$ for all $x, y \in \mathcal{X}$. As we further discuss in Appendix [C](#), the Bregman divergence measures how far two points are in a geometry induced by the mirror map. Given a starting policy $\pi^0$, a learning rate $\eta$ and a mirror map $h$, PMD can be formalized as an iterative algorithm: for all iterations $t \geq 0$,

$$\pi^{t+1} \in \operatorname{argmax}_{\pi \in (\Delta(\mathcal{A}))^{\mathcal{S}}} \mathbb{E}_{s \sim d_\mu^t}[\eta_t \langle Q_s^t, \pi_s \rangle - \mathcal{D}_h(\pi_s, \pi_s^t)], \tag{2}$$

where we used the shorthand: $V^t := V^{\pi^t}$, $Q^t := Q^{\pi^t}$, $d_\mu^t := d_\mu^{\pi^t}$ and $y_s := y(s, \cdot) \in \mathbb{R}^{\mathcal{A}}$, for any function $y : \mathcal{S} \times \mathcal{A} \to \mathbb{R}$. PMD benefits from several theoretical guarantees and, in particular, [Xiao](#) ([2022](#)) shows that PMD enjoys quasi-monotonic updates and convergence to the optimal policy. We give here a slight modification of the statements of these results. Specifically, we do not upper-bound a term regarding the distance between subsequent policies in the result on quasi-monotonic updates, which we use to draw connections between theory and practice. Additionally, to obtain the convergence rate for PMD with constant step-size in the setting where we do not have access to the true Q-function, we combine the analyses on the sublinear convergence of PMD and linear convergence of inexact PMD given by [Xiao](#) ([2022](#)). We provide a proof in Appendix [D](#).

**Theorem 2.2** ([Xiao](#) ([2022](#))). *Following update* ([2](#)), *we have that, for all* $t \geq 0$

$$V^{t+1}(\mu) - V^t(\mu) \geq -\frac{1}{1-\gamma} \max_{s \in \mathcal{S}} \|\widehat{Q}_s^t - Q_s^t\|_\infty \max_{s \in \mathcal{S}} \|\pi_s^{t+1} - \pi_s^t\|_1, \tag{3}$$

*where* $\|\cdot\|_\infty$ *and* $\|\cdot\|_1$ *represent the* $\ell_\infty$ *and the* $\ell_1$ *norms, respectively, and* $\widehat{Q}$ *is an estimate of the true Q-function. Additionally, at each iteration* $T > 0$, *we have*

$$V^\star(\mu) - \sum_{t<T} \frac{\mathbb{E}[V^t(\mu)]}{T} \leq \frac{1}{T} \left( \frac{\mathbb{E}_{s \sim d_\mu^\star}[\mathcal{D}_h(\pi_s^\star, \pi_s^0)]}{\eta_t(1-\gamma)} + \frac{1}{(1-\gamma)^2} \right) + 4 \frac{\max_{t<T, s \in \mathcal{S}} \|\widehat{Q}_s^t - Q_s^t\|_\infty}{(1-\gamma)^2}. \tag{4}$$

The statement of Theorem [2.2](#) is similar to many results in the literature on PMD and NPG ([Agarwal et al., 2021](#); [Xiao, 2022](#); [Lan, 2022b](#); [Hu et al., 2022](#)). That is, the convergence guarantee involves two terms, i.e. a convergence rate, which involves the Bregman divergence between the optimal policy and the starting policy, and an error floor, which involves the estimation error $\max_{s \in \mathcal{S}} \|\widehat{Q}_s^t - Q_s^t\|_\infty$. Given that by setting $\eta_0 = \mathbb{E}_{s \sim d_\mu^\star}[\mathcal{D}_h(\pi_s^\star, \pi_s^0)](1-\gamma)$ we obtain the convergence rate $2(T(1-\gamma)^2)^{-1}$, and that the error floor has no explicit dependence on the mirror map, Equation ([4](#)) suggests that the mirror map has a mild influence on the performance of PMD. The only way the mirror map seems to affect the convergence guarantee is, implicitly, by changing the path of the algorithm and therefore influencing the estimation error. Similar observations can be made for several results in the PMD literature that share the same structure of convergence guarantees. On the other hand, Equation ([3](#)) suggests that mirror maps that prevent large updates of the policy cause the PMD algorithm to be less prone to performance degradation, as the lower bound is close to 0 when the policy update distance $\max_{s \in \mathcal{S}} \|\pi_s^{t+1} - \pi_s^t\|_1$ is small. One of the contributions of

our work is to challenge and refute the conclusion that the mirror map has little influence on the convergence of PMD, highlighting a gap between theoretical guarantees, which are *based on upper bounds*, and the actual performance observed in PMD-based methodologies. Our empirical studies reveal that the choice of mirror map significantly influences both the speed of convergence and the minimum achievable error floor in PMD. Additionally, we provide evidence that a mirror map that prevents large updates throughout training leads to a better performance, as suggested by (3).

When applying PMD to continuous control tasks, we replace the tabular policy in (2) with a parametrized one. That is, for all iterations $t \geq 0$, we obtain the updated policy as

$$\pi_{\theta^{t+1}} \in \mathrm{argmax}_{\pi_\theta : \theta \in \Theta} \, \mathbb{E}_{s \sim d_\mu^t}[\eta_t \mathbb{E}_{a \sim \pi_\theta(\cdot|s)}(Q^t(s,a)) - \mathcal{D}_h(\pi_\theta(\cdot|s), \pi^t(\cdot|s))], \qquad (5)$$

where $\{\pi_\theta : \theta \in \Theta\}$ is a class of parametrized policies.

## 2.3 Approximate Mirror Policy Optimization

AMPO is a theoretically sound framework for deep reinforcement learning based on mirror descent, as it inherits the quasi-monotonic updates and convergence guarantees from the tabular case (Alfano et al., 2023). Given a parameterized function class $\mathcal{F}^\Theta = \{f^\theta : \mathcal{S} \times \mathcal{A} \to \mathbb{R}, \theta \in \Theta\}$, an initial scoring function $f^{\theta^0}$, a step-size $\eta$ and an $\omega$-potential mirror map $h_\phi$, AMPO can be described, for all iterations $t$, by the two-step update

$$\pi^t(a \mid s) = \sigma(\phi(\eta f^t(s,a) + \lambda_s^t)) \qquad \forall s \in \mathcal{S}, a \in \mathcal{A}, \qquad (6)$$

$$\theta^{t+1} \in \underset{\theta \in \Theta}{\mathrm{argmin}} \, \mathbb{E}_{s \sim d_\mu^t, a \sim \pi(\cdot|s)} \left[ \left( f^\theta(s,a) - Q^t(s,a) - \eta^{-1} \max(\eta f^t(s,a) + \lambda_s^t, \phi^{-1}(0)) \right)^2 \right] \quad (7)$$

where $\lambda_s^t \in \mathbb{R}$ is a normalization factor to ensure $\pi_s^t \in \Delta(\mathcal{A})$ for all $s \in \mathcal{S}$, $f^t := f^{\theta^t}$, and $\sigma(z) = \max(z, 0)$ for $z \in \mathbb{R}$. Theorem 1 by Krichene et al. (2015) ensures that there always exists a normalization constant $\lambda_s^t \in \mathbb{R}$. As shown by Alfano et al. (2023), AMPO recovers the standard formulation of PMD in (2) in the tabular setting. Assuming for simplicity that $\phi(x) > 0$ for all $x \in \mathbb{R}$, the minimization problem in (7) implies that, at each iteration $t$, $f^t$ is an approximation of the sum of the $Q$-functions up to that point, that is $f^t \simeq \sum_{i=0}^{t-1} Q^i$. Therefore, the scoring function $f^t$ serves as an estimator of the value of an action.

## 3 Methodology

Denote by $\mathcal{H}$ the class of $\omega$-potentials mirror maps. Our objective is to search for the mirror map that maximizes the value of the last policy outputted by our mirror descent based algorithms, for a fixed time horizon $T$. That is, we want to find

$$h^\star \in \underset{h \in \mathcal{H}}{\mathrm{argmax}} \, \mathbb{E}\left[V^T(\mu)\right], \qquad (8)$$

where the expectation is taken over the randomness of the policy optimization algorithm and the policy updates are based on the mirror map $h$. Depending on the setting, we parameterize the mirror map by parameterizing either $\phi^{-1}$ or $\phi$ as monotonically increasing functions. One of the primary objectives of this work it to motivate further research into the choice of mirror maps, as influenced by the choice of the function $\phi$, moving beyond the conventional use of $\phi(x) = e^x$. This is achieved by examining how different choices of $\phi$ influence the trajectory of training and demonstrating that, in many cases, there is a mirror map that outperforms the negative entropy by a large margin.

### 3.1 Policy Mirror Descent

We parameterize $\phi^{-1}$ as a one layer neural network with 126 hiden units, where all kernels are non-negative and the activation functions are equally split among the following convex and concave monotonic non-linearities: $x^3$, $(x)_+^2$, $(x)_+^{1/2}$, $(x)_+^{1/3}$, $\log((x)_+ + 10^{-3})$ and $e^x$, where $(x)_+ = \max(x, 0)$. To ensure that we are able to recover the negative entropy and the $\ell_2$-norm, we add $ax + b\log(x)$ to the final output, where $a, b \geq 0$

To search for the best mirror map within this class, we employ a slight variation of the OpenAI-ES strategy (Salimans et al., 2017), adapted to the multi-task setting (Jackson et al., 2024). Denote by $\psi$ the parameters of the mirror map and by $F(\psi)$ the objective function in (8). Given a distribution of tasks $\mathcal{E}$, we estimate the gradient $\nabla_\psi F(\psi)$ as

$$E_{\epsilon \sim \mathcal{N}(0,I_d)} \left[ E_{e \sim \mathcal{E}} \left[ \frac{\epsilon}{2\sigma} (F_e(\psi + \sigma\epsilon) - F_e(\psi - \sigma\epsilon)) \right] \right], \tag{9}$$

where $\mathcal{N}(0, I_d)$ is the multivariate normal distribution, $d$ is the number of parameters, and $\sigma > 0$ is a hyperparameter regulating the variance of the perturbations. To account for different reward scales across tasks, we perform a rank transformation of the objective functions, whereby, for each sampled task $e$, we return 1 for the higher performing member between $\psi + \sigma\epsilon$ and $\psi - \sigma\epsilon$, and 0 for the other. We note that, when the distribution of tasks $\mathcal{E}$ covers a single task, we recover the standard OpenAI-ES strategy.

## 3.2 APPROXIMATE MIRROR POLICY OPTIMIZATION

As non-tabular environments with discrete action space, we consider the Basic Control Suite (BCS) and the MinAtar suite. Given the higher computational cost of simulations on these environments w.r.t. to the tabular setting, we replace the neural network parameterization for the mirror map with one with fewer parameters, in order to reduce the dimension of the mirror map class we search over. We define the parameterized class $\Phi = \{\phi_\psi : \mathbb{R} \to [0,1], \psi \in \mathbb{R}^n_+\}$, with

$$\phi_\psi(x) = \begin{cases} 0 & \text{if } x \leq 0, \\ \frac{x}{\psi_1 n} & \text{if } 0 < x \leq \psi_1, \\ \frac{j}{n} + \frac{x - \sum_{i=1}^j \psi_i}{n\psi_{j+1}} & \text{if } \sum_{i=1}^j \psi_i < x \leq \sum_{i=1}^{j+1} \psi_i, \\ 1 & \text{if } x > 1, \end{cases}$$

where $1 \leq j \leq n - 1$. In other words, $\phi$ is defined as a piece-wise linear function with $n$ steps where, for all $j \leq n$, $\phi(\sum_{i=1}^j \psi_i) = j/n$ and subsequent points are interpolated with a straight segment. This is illustrated in Figure 1. We note that the $\omega$-potentials within $\Phi$ violate some of the constraints in Definition 2.1, as they are non-decreasing instead of increasing and $\lim_{x\to\infty} \phi(x) = 1$ for all $\phi \in \Phi$. In Appendix B.2, we show that, if $\phi \in \Phi$, we can construct an $\omega$-potential $\phi'$ that satisfies the constraints in Definition 2.1 such that $\phi$ and $\phi'$ induce the same policies along the path of AMPO.

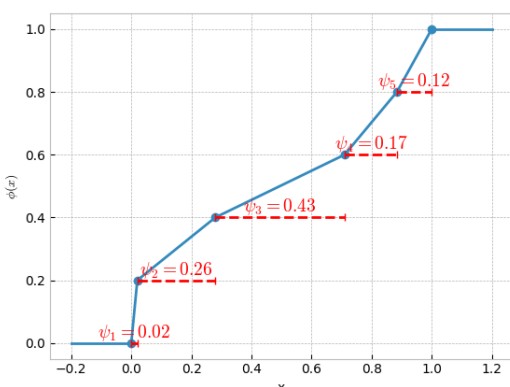

Figure 1: A plot visually demonstrating the parameterization for $\phi$.

To effectively learn the hyperparameters, we employ the Separable Covariance Matrix Adaptation Evolution Strategy (sep-CMA) (Ros & Hansen, 2008), a variant of the popular algorithm CMA-ES (Hansen & Ostermeier, 2001). CMA-ES is, essentially, a second-order method adapted for gradient free optimization. At every generation, it samples $n$ new points from a normal distribution, parameterized by a mean vector $m_k$ and a covariance matrix $C_k$. That is, the samples for generation $k$, $x_1^k, \ldots, x_n^k$, are distributed i.i.d. according to $x_i^k \sim \mathcal{N}(m_k, C_k)$. For each generation $k$, denote the $m \leq n$ best performing samples as $x_1^{*k}, \ldots, x_m^{*k}$. Then update the mean vector as $m_{k+1} = \sum_{i=1}^m w_i x_i^{*k}$, where $\sum_{i=1}^m w_i = 1$, so that the next generation is distributed around the weighted mean of the best performing samples. $C_{k+1}$ is also updated to reflect the covariance structure of $x_1^{*k}, \ldots, x_m^{*k}$, in a complex way beyond the scope of this text. Due to the need to update $C_k$ based on covariance information, CMA-ES exhibits a quadratic scaling behavior with respect to the dimensionality of the search space, potentially hindering its efficiency in high-dimensional settings. To improve computational efficiency, we adopt sep-CMA, which introduces a diagonal constraint on the covariance matrix and reduces the computational complexity of the algorithm.

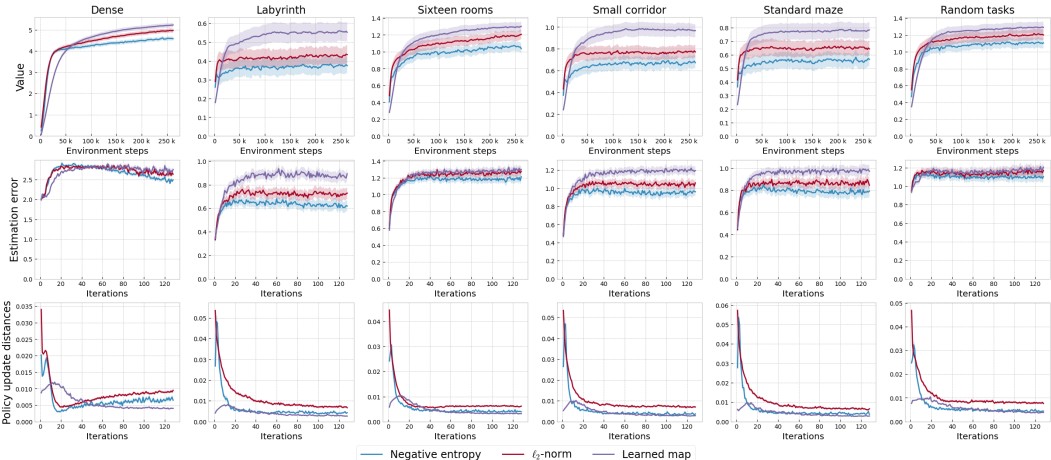

Figure 2: Comparison between the learned map and the negative entropy and $\ell_2$-norm mirror maps across a range of held-out configurations of Grid-World. We display the average over 256 runs and report the standard error as a shaded region. The column "Random tasks", reports the averaged metrics for 256 randomly sampled configurations of Grid-World.

## 4 EXPERIMENTS

In this section, we discuss the results of our numerical experiments. We start by presenting the tabular setting, where we track errors in order to understand what properties are desirable in a mirror map, and proceed by showing our results in the non-tabular setting.

### 4.1 TABULAR SETTING: GRID-WORLD

As tabular setting we adopt a discounted and infinite horizon version of Grid-World (Oh et al., 2020), which is a large class of tabular MDPs.

**Model architecture and training** We define the tabular policy as a one-layer neural network with a softmax head, which takes as input a one-hot encoding of the environment state and outputs a distribution over the action space. We train the policy using the PMD update in (2), where we solve the minimization problem through stochastic gradient descent and estimate the $Q$-function through generalized advantage estimation (GAE) (Schulman et al., 2016). We perform a simple grid-search over the hyperparameters to maximize the performance for the negative entropy and the $\ell_2$-norm mirror maps, in order to have a fair comparison. We report the chosen hyperparameters in Appendix E. The training procedure is implemented in Jax, using evosax (Lange, 2022a) for the evolution. We run on four A40 GPUs, and the optimization process takes roughly 12 hours.

**Environment** We adopt the version of Grid-World implemented by Jackson et al. (2024), and adapt it to the discounted and infinite horizon setting. We learn a single mirror map by training PMD on a continuous distribution of Grid-World environments, and test PMD with the learned mirror map on five held-out configurations from previous publications (Oh et al., 2020; Chevalier-Boisvert et al., 2024) and on 256 randomly sampled configurations. For all PMD iterations $t$, we track two quantities that appear in Theorem 2.2, that is the estimation error $\max_{s \in \mathcal{S}} \|\widehat{Q}_s^t - Q_s^t\|_\infty$, and the distance between policy updates $\max_{s \in \mathcal{S}} \|\pi_s^{t+1} - \pi_s^t\|_1$. To obtain the true $Q$-function, we compute the transition matrix $P^{\pi^t}$ and use the formula specified in Section 2, that is $Q^t = (I - \gamma P^{\pi^t})^{-1} r$. PMD is run for 128 iterations and $2^{18} \simeq 250k$ total environment steps for all Grid-World configurations.

We show the results of our simulations in Figure 2. As shown by the first row, the learned mirror map, the $\ell_2$-norm, and the negative entropy consistently rank first, second and third, respectively, in all tested configurations, in terms of final value. These results advocate the effectiveness of our methodology, as we are able to find a mirror map that outperforms the benchmark mirror maps in all tested environments. We highlight that the first five environments in Figure 2 are not part of the task

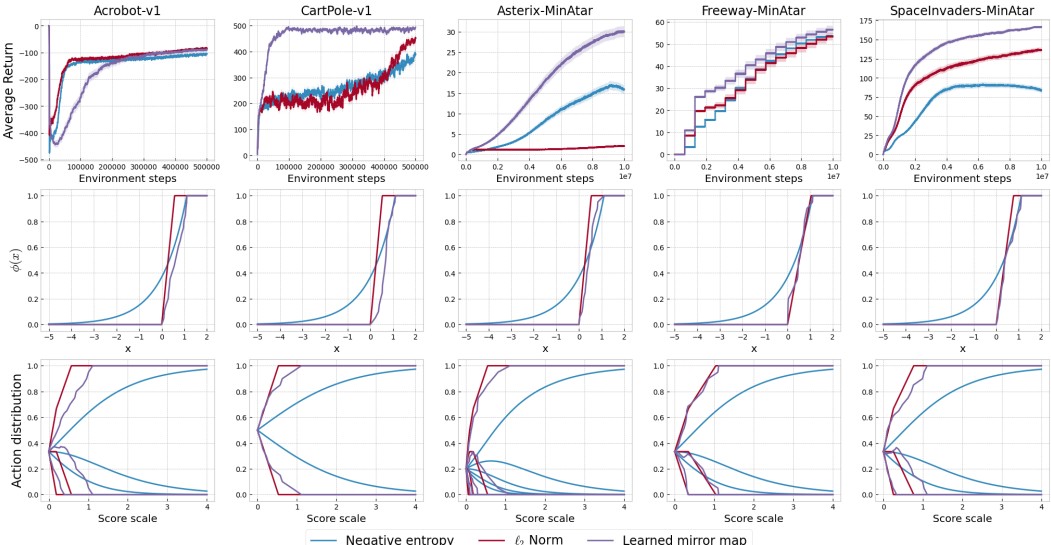

Figure 3: Comparison between the learned mirror map, the $\ell_2$-norm and the negative entropy across a range of standard environments. The top plots present the performance of AMPO for all mirror maps, reporting the average over 100 realizations and a shaded region denoting the standard error around the average. The middle plots report the $\omega$-potentials that induce the mirror maps. The bottom plots report the policy distribution according to (6), for each mirror map and score scales. The score scales are obtained by multiplying the vector $[1, \ldots, |\mathcal{A}|]$ by a variable $c \in [0, 4]$.

distribution used during the evolution, demonstrating the generalizability of our approach. Moreover, we have that the $\ell_2$-norm consistently outperforms the negative entropy, further proving that the negative entropy is not always the best choice of mirror map. Another shared property among all training curves, is that the learned mirror map presents a slower convergence in the initial iterations w.r.t. the $\ell_2$-norm and to the negative entropy, as testified by a lower value, but convergence to a higher value in the long run. Lastly, we note that in all environments most of the value improvement happens in the first 50k environment steps.

To gain a better understanding of why the learned mirror maps outperforms the negative entropy and the $\ell_2$-norm, and to draw connections with the theoretical results outlined in Section 2.2, we report the estimation error and the distance between policy updates for all mirror maps. The first conclusion that we draw is that a smaller estimation error does not seem to be related to a higher performance. On the contrary, the second row of Figure 2 shows how the three mirror maps consistently have the same ranking in both value and estimation error, which is exactly the opposite of what Theorem 2.2 would suggest. On the other hand, the first and third rows of Figure 2 show how the lower bound on the performance improvement in (3) brings some valid insight on the behaviour of PMD during its first iterations, which are the ones that bring the largest improvement. In all configurations, we have that in the initial iterations of PMD the learned mirror map induces the smallest policy update distances as well as the performance curve with fewer dips in value, while both the negative entropy and the $\ell_2$-norm induce larger policy update distances and performance curves with several dips in value. This observation confirms the behaviour described by (3), whereby a small distance between policy updates prevents large performance degradation in policy updates.

## 4.2 Non-tabular setting: Basic Control and MinAtar suites

**Model architecture and training** We define the scoring function as a deep neural network, which we train using the AMPO update in (7) and (6), where (7) is solved through Adam and the $Q$-function is estimated through GAE. We optimize the hyper-parameters of AMPO for the negative entropy mirror map for each suite, using the hyper-parameter tuning framework Optuna (Akiba et al., 2019). This is done to ensure we are looking at a fair benchmark of performance when using the negative entropy mirror map. We report the chosen hyper-parameters in Appendix E. We then initialize the

Table 1: The table contains, for each entry, the value of the final policy outputted by AMPO trained on the environment corresponding to the column with the mirror map learned on the environment corresponding to the row. The last row represents the performance of AMPO with the negative entropy for the corresponding column environments. The value is averaged over 100 runs. Green cells correspond to a value higher than that associated to the negative entropy.

| | Acrobot | CartPole | Asterix | Freeway | SpaceInvaders |
|---|---|---|---|---|---|
| Acrobot | -88.49 | 476.41 | 24.03 | 56.00 | 144.01 |
| CartPole | -83.76 | 499.93 | 27.26 | 52.26 | 100.07 |
| Asterix | -103.55 | 490.86 | 30.22 | 58.56 | 122.00 |
| Freeway | -82.51 | 457.47 | 3.20 | 58.21 | 143.93 |
| SpaceInvaders | -78.29 | 489.56 | 4.36 | 22.27 | 170.24 |
| Negative entropy | -105.63 | 359.14 | 17.80 | 53.69 | 81.77 |

parameterized mirror map to be an approximation of the negative entropy[2] and run Sep-CMA-ES on each environment separately. The whole training procedure is implemented in JAX, using gymnax environments (Lange, 2022b) and evosax (Lange, 2022a) for the evolution. We run on 8 GTX 1080Ti GPUs, and the optimization process takes roughly 48 hours for a single environment.

**Environments**  We test AMPO on the Basic Controle Suite (BCS) and the MinAtar Suite. For BCS, we run the evolution for 600 generation, each with 500k timesteps. For MinAtar, we run the evolution for 500 generation with 1M timesteps, then run 100 more generations with 10M timesteps.

Our empirical results are illustrated in Figure 3, where we show the performance of AMPO for the learned mirror map, for the negative entropy and for the $\ell_2$-norm. For all environments and mirror maps, we use the hyper-parameters returned by Optuna for AMPO with the negative entropy. Our learned mirror map leads to a better overall performance in all environments, apart from Acrobot, where it ties with the $\ell_2$-norm. We observe the largest improvement in performance on Asterix and SpaceInvaders, where the average return for the learned mirror map is more than double the one for the negative entropy. Figure 3 also suggests that different mirror maps may result in different error floors, as shown by the performance curves in Acrobot, Asterix and SpaceInvaders, where the negative entropy converges to a lower point than the learned mirror map.

The second and third row of Figure 3 illustrate the properties of the learned mirror map, in comparison to the negative entropy and the $\ell_2$-norm. In particular, the second row shows the corresponding $\omega$-potential, while the third row shows the policy distribution induced by the mirror map according to (6), depending on the scores assigned by the scoring function to each action. A shared property among all the learned mirror maps is that they all lead to assigning $0$ probability to the worst actions for relatively small score scales, whilst the negative entropy always assigns positive weights to all actions. In more complex environments, where the evaluation of a certain action may be strongly affected by noise or where the optimal state may be combination locked (Misra et al., 2020), this behaviour may lead to a critical lack of exploration. However, it appears that this is not the case in these environments, and we hypothesise that by setting the probability of the worst actions to $0$ the learned mirror maps avoid wasting samples and hence can converge to the optimal policy more rapidly.

Our last result consists in testing each learned mirror map across the other environments we consider. In Table 1, we report the value of the final policy outputted by AMPO for all learned maps, plus the negative entropy, and for all environments, averaged over 100 runs. The table shows that the learned mirror maps generalize well to different environments and to different sets of hyper-parameters, which are shared within BCS and MinAtar but not between them. In particular, we have that the mirror maps learned in Acrobot and Asterix outperform the negative entropy in all environments, those learned in CartPole and Freeway outperform the negative entropy in 4 out of 5 environments, and that learned in SpaceInvaders outperforms the negative entropy in 3 out of 5 environments. These results show that our methodology can be useful in practice, as it benefits from good generalization across tasks.

---

[2]We achieve this by assigning $\psi_i \propto \log(i/(i-1))$, for $i = 2, \dots, n$, and setting $\psi_1 \propto 3\log(10)$.

### 4.3 Continuous Control: MuJoCo

**Model architecture and training**   We define the policy as a deep neural network and optimize it using the PMD update in (5), which is computed using Adam and the estimate of the $Q$-function obtained through GAE. As previously discussed, we ensure a fair comparison by tuning the hyper-paramers to maximize the performance of the negative entropy. We report the chosen hyper-parameters in Appendix E. The mirror map is initialized to be close to the negative entropy and is trained for 256 generations using OpenAI-ES on a single MuJoCo environment. We run on eight A40 GPUs, and the optimization process takes roughly 24 hours.

**Environments**   We use the `brax` library (Freeman et al., 2021) to simulate three MuJoCo environments, i.e. Hopper, Halfcheetah, and Ant. The mirror map is learned on Hopper and is then tested on all environments, using $10^7$ environment steps and 488 PMD update steps.

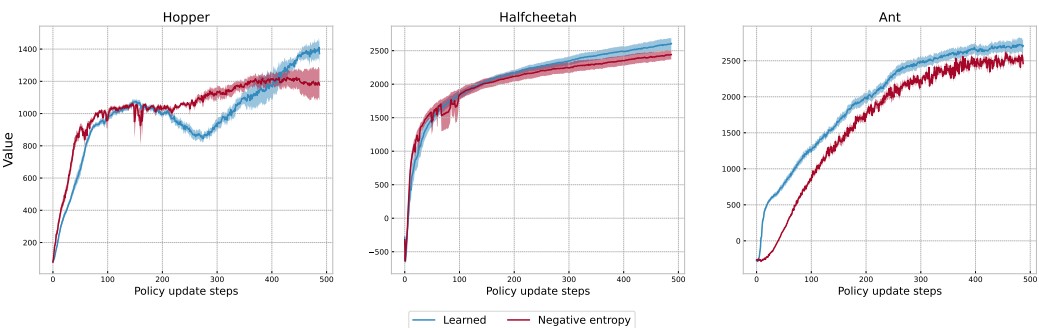

Figure 4: Comparison between the mirror map learned on Hopper and the negative entropy on three MuJoCo environments. The plots present the performance of PMD for both mirror maps, reporting the average over 8 realizations and a shaded region denoting the standard error around the average.

Figure 4 shows that the mirror map learned on Hopper significantly improves in terms of final performance upon the negative entropy in both train and test environments. This result confirms the ability of our methodology to learn generalizable mirror maps, even in complex continuous control tasks.

## 5 Conclusion

Our study presents an empirical examination of PMD, where we successfully test the possibility of learning a mirror map that outperforms the negative entropy in both the tabular and non-tabular settings. In particular, we have shown that the learned mirror maps perform well on a set of configurations in Grid-World and that they can generalize to different tasks in BCS, in MinAtar, and in MuJoCo. Additionally, we have compared the theoretical findings established in the literature and the actual performance of PMD methods in the tabular setting, highlighting how the estimation error is not a good indicator of performance and validating the intuition that small policy updates lead to less instances of performance degradation. Our findings indicate that the choice of mirror map significantly impacts PMD's effectiveness, an aspect not adequately reflected by existing convergence guarantees.

Our research introduces several new directions for inquiry. From a theoretical perspective, obtaining convergence guarantees that reflect the impact of the mirror map is an area for future exploration. On the practical side, investigating how temporal awareness (Jackson et al., 2024) or specific environmental challenges, such as exploration, robustness to noise, or credit assignment (Osband et al., 2019), can inform the choice of a mirror map to improve performance represents another research focus.

#### Acknowledgments

Carlo Alfano is funded by the Engineering and Physical Sciences Research Council. Patrick Rebeschini was funded by UK Research and Innovation (UKRI) under the UK government's Horizon Europe funding guarantee [grant number EP/Y028333/1]. For the purpose of Open Access, the authors have applied a CC BY public copyright licence to any Author Accepted Manuscript (AAM) version arising from this submission.

REPRODUCIBILITY STATEMENT

The implementation of our experiments can be found at https://github.com/c-alfano/Learning-mirror-maps.

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

# A  RELATED WORKS

Discovering reinforcement learning algorithm has been an active area of research in the last years, where researchers have shown that handcrafted algorithms are not always optimal and can be outperformed by automatically discovered algorithms. Oh et al. (2020) obtained an algorithm capable of solving GridWorld and Atari tasks without relying on common notions in the field, such as Q-value functions. Houthooft et al. (2018) used ES to meta-train a policy loss network that outperforms PPO and Kirsch et al. (2020) discovered a new loss function for deterministic policies. Most closely related to our work are Lu et al. (2022) and Jackson et al. (2024), who employ ES to discover RL algorithms within the Mirror Learning class (Kuba et al., 2022). Given a function $f$, the class of algorithms they consider follows the policy update step

$$\pi_{\theta^{t+1}} \in \text{argmax}_{\pi_\theta : \theta \in \Theta} \, \mathbb{E}_{s \sim d_\mu^t} \left[ \eta_t \mathbb{E}_{a \sim \pi_\theta(\cdot|s)}(Q^t(s,a)) - \mathbb{E}_{a \sim \pi^t(\cdot|s)} \left[ \left( \frac{\pi_\theta(\cdot|s)}{\pi^t(\cdot|s)} \right) \right] \right],$$

This update is similar the the one in (5) but replaces the Bregman divergence with a different penalty term called *drift*, which recovers the $f$-divergences when $f$ is a convex function. To the best of our knowledge, this class of algorithms has fewer theoretical guarantees than PMD, in particular we are not aware of finite-time convergence guarantees or convergence guarantees involving approximation or estimation errors (Belousov & Peters, 2017; Kuba et al., 2022).

# B  FURTHER DISCUSSION ON $\omega$-POTENTIALS

## B.1  NEGATIVE ENTROPY AND $\ell_2$-NORM

If $\phi(x) = e^{x-1}$, then the associated mirror map $h_\phi$ is the negative entropy. We have that

$$\int_0^1 \phi^{-1}(x)dx = \int_0^1 \log(x)dx = [x\log(x) - x]_0^1 = -1 \leq +\infty.$$

The mirror map $h_\phi$ becomes the negative entropy, up to a constant, as

$$h_\phi(\pi_s) = \sum_{a \in \mathcal{A}} \int_1^{\pi(a|s)} \log(x)dx = |\mathcal{A}| - 1 + \sum_{a \in \mathcal{A}} \pi(a \mid s)\log(\pi(a \mid s)).$$

If $\phi(x) = x$, then the associated mirror map $h_\phi$ is the $\ell_2$-norm. We have that

$$\int_0^1 \phi^{-1}(x)dx = \int_0^1 xdx = \left[\frac{x^2}{2}\right]_0^1 = \frac{1}{2} \leq +\infty.$$

The mirror map $h_\phi$ becomes the $\ell_2$-norm, up to a constant, as

$$h_\phi(\pi_s) = \sum_{a \in \mathcal{A}} \int_1^{\pi(a|s)} xdx = \frac{1}{2} \sum_{a \in \mathcal{A}} \pi(a \mid s)^2 - 1.$$

## B.2  PARAMETRIC $\phi$

We show here that the parametric class of $\omega$-potentials we introduce in Section 3 results in a well defined algorithm when used for AMPO, even if it breaks some of the constraints in Definition 2.1. In particular, $\omega$-potentials within $\Phi$ are not $C^1$-diffeomorphisms and are only non-decreasing. We can afford not meeting the first constraint as the proof for Theorem 1 by Krichene et al. (2015), which establishes the existence of the normalization constant in Equation (6), only requires Lipschitz-continuity. As to the second constraint, we build an augmented class $\Phi'$ that contains increasing $\omega$-potentials and show that it results in the same updates for AMPO as $\Phi$.

Let $\Delta_n$ be the $n$-dimensional probability simplex. We define the parameterized class $\Phi' = \{\phi'_\psi : \mathbb{R} \to [0,1], \psi \in \Delta_n\}$, with

$$\phi'_\psi(x) = \begin{cases} e^x - 1 & \text{if } x \leq 0, \\ \frac{x}{\psi_1 n} & \text{if } 0 < x \leq \psi_1, \\ \frac{j}{n} + \frac{x - \sum_{i=1}^j \psi_i}{n\psi_{j+1}} & \text{if } \sum_{i=1}^j \psi_i < x \leq \sum_{i=1}^{j+1} \psi_i, \\ x & \text{if } x > 1, \end{cases}$$

where $1 \leq j \leq n-1$. The functions within $\Phi'$ are increasing and continuous. We start by showing that $\phi'_\psi \in \Phi'$ and $\phi_\psi \in \Phi$ are equivalent for Equation (6), for the same parameters $\psi$. At each iteration $t$, we have that

$$\pi^{t+1}(a \mid s) = \sigma(\phi'(\eta f^{t+1}(s,a) + (\lambda')_s^{t+1}))$$
$$= \sigma(\phi(\eta f^{t+1}(s,a) + (\lambda')_s^{t+1}))$$
$$= \sigma(\phi(\eta f^{t+1}(s,a) + \lambda_s^{t+1})).$$

This due to the following two facts. For $x \leq 0$, $\sigma(\phi'(x))$ and $\sigma(\phi(x))$ are the same function. Also, $\sigma(\phi'(\eta f^{t+1}(s,a) + (\lambda')_s^{t+1}))$ has to be less or equal to 1, as a result of the projection, meaning that $\eta f^{t+1}(s,a) + (\lambda')_s^{t+1} \leq 1$, where $\sigma(\phi'(\cdot))$ and $\sigma(\phi(\cdot))$ are equivalent.

Since $\phi$ and $\phi'$ induce the same policy, share that same normalization constant, and have the same value at 0, they also induce the same expression for Equation (7).

## C   FURTHER DISCUSSION ON BREGMAN DIVERGENCE

To provide an intuition on the Bregman divergence, we report here some discussion from Orabona (2020). Given a mirror map $h$, the associated Bregman divergence between two points $x, y \in \mathcal{X}$ is defined as

$$\mathcal{D}_h(x,y) := h(x) - h(y) - \langle \nabla h(y), x - y \rangle.$$

Since $h$ is convex, the Bregman divergence is always non-negative for $x, y \in \mathcal{X}$. We can build more intuition for the Bregman divergence using Taylor's theorem. Assume that $h$ is twice differentiable in an open ball $B$ around $y$ and $x \in B$. Then there exists $0 \leq \alpha \leq 1$ such that

$$B_\psi(x; y) = h(x) - h(y) - \nabla h(y)^\top (x - y) = \frac{1}{2}(x-y)^\top \nabla^2 h(z)(x-y),$$

where $z = \alpha x + (1-\alpha)y$. Therefore, the Bregman divergence can be approximated by squared local norms that depend on the Hessian of the mirror map $h$. This means that the Bregman divergence will behave differently on different areas of $\mathcal{X}$, depending on the value of Hessian of the mirror map.

## D   PROOF OF THEOREM 2.2

In this section we outline how we use the proofs given by Xiao (2022) in order to obtain Theorem 2.2.

### D.1   PRELIMINARY LEMMAS

We start by presenting two preliminary lemmas, which are ubiquitous in the literature on PMD. The first characterizes the difference in value between two policies, while the second characterizes the PMD update.

**Lemma D.1** (Performance difference lemma, Lemma 1 in Xiao (2022)). *For any policy $\pi, \pi' \in \Delta(\mathcal{A})^{\mathcal{S}}$ and $\mu \in \Delta(\mathcal{S})$,*

$$V^\pi(\mu) - V^{\pi'}(\mu) = \frac{1}{1-\gamma} \mathbb{E}_{s \sim d_\mu^\pi} \left[ \langle Q_s^{\pi'}, \pi_s - \pi'_s \rangle \right].$$

**Lemma D.2** (Three-point decent lemma, Lemma 6 in Xiao (2022)). *Suppose that $\mathcal{C} \subset \mathbb{R}^m$ is a closed convex set, $f : \mathcal{C} \to \mathbb{R}$ is a proper, closed [3] convex function, $\mathcal{D}_h(\cdot, \cdot)$ is the Bregman divergence generated by a mirror map $h$. Denote $\mathrm{rint}\,\mathrm{dom}\,h$ as the relative interior of $\mathrm{dom}\,h$. For any $x \in \mathrm{rint}\,\mathrm{dom}\,h$, let*

$$x^+ \in \arg\min_{u \in \mathrm{dom}\,h \cap \mathcal{C}} \{f(u) + \mathcal{D}_h(u,x)\}.$$

*Then $x^+ \in \mathrm{rint}\,\mathrm{dom}\,h \cap \mathcal{C}$ and for any $u \in \mathrm{dom}\,h \cap \mathcal{C}$,*

$$f(x^+) + \mathcal{D}_h(x^+, x) \leq f(u) + \mathcal{D}_h(u,x) - \mathcal{D}_h(u, x^+).$$

---

[3]A convex function $f$ is proper if $\mathrm{dom}\,f$ is nonempty and for all $x \in \mathrm{dom}\,f$, $f(x) > -\infty$. A convex function is closed, if it is lower semi-continuous.

## D.2 QUASI-MONOTONIC UPDATES

We first prove that PMD enjoys quasi-monotonic updates (Equation (3)), that is PMD updates have an upper bound on how much they can deteriorate performance.

**Proposition D.3** (Lemma 11 in Xiao (2022))**.** *At each time $t \geq 0$, we have*

$$\langle \eta_t \widehat{Q}_s^t, \pi_s^{t+1} - \pi_s^t \rangle \geq 0.$$

*Additionally, we have that*

$$V^{t+1}(\mu) - V^t(\mu) \geq -\frac{1}{1-\gamma} \max_{s \in \mathcal{S}} \left\| \widehat{Q}_s^t - Q_s^t \right\|_\infty \max_{s \in \mathcal{S}} \left\| \pi_s^{t+1} - \pi_s^t \right\|_1. \tag{10}$$

*Proof.* Using Lemma D.2 with $x^+ = \pi_s^{t+1}$, $\mathcal{C} = \Delta(\mathcal{A})$, $f(u) = \langle \widehat{Q}_s^t, u \rangle$, $x = \pi^t$ and $u = \pi^{t+1}$, we obtain

$$\langle \eta_t \widehat{Q}_s^t, \pi_s^t - \pi_s^{t+1} \rangle \leq \mathcal{D}_h(\pi_s^t, \pi_s^t) - \mathcal{D}_h(\pi_s^{t+1}, \pi_s^t) - \mathcal{D}_h(\pi_s^t, \pi_s^{t+1}). \tag{11}$$

By rearranging terms and noticing $\mathcal{D}_h(\pi_s^t, \pi_s^t) = 0$, we have

$$\langle \eta_t \widehat{Q}_s^t, \pi_s^{t+1} - \pi_s^t \rangle \geq \mathcal{D}_h(\pi_s^{t+1}, \pi_s^t) + \mathcal{D}_h(\pi_s^t, \pi_s^{t+1}) \geq 0. \tag{12}$$

Equation (10) can be obtained using the performance difference lemma and (12):

$$\begin{aligned}
V^{t+1}(\mu) - V^t(\mu) &= \frac{1}{1-\gamma} \mathbb{E}_{s \sim d_\mu^{t+1}} \left[ \langle Q_s^t, \pi_s^{t+1} - \pi_s^t \rangle \right] \\
&= \frac{1}{1-\gamma} \mathbb{E}_{s \sim d_\mu^{t+1}} \left[ \langle \widehat{Q}_s^t, \pi_s^{t+1} - \pi_s^t \rangle \right] \\
&\quad + \frac{1}{1-\gamma} \mathbb{E}_{s \sim d_\mu^{t+1}} \left[ \langle \widehat{Q}_s^t - Q_s^t, \pi_s^{t+1} - \pi_s^t \rangle \right] \\
&\geq -\frac{1}{1-\gamma} \mathbb{E}_{s \sim d_\mu^{t+1}} \left[ \left\| \widehat{Q}_s^t - Q_s^t \right\|_\infty \left\| \pi_s^{t+1} - \pi_s^t \right\|_1 \right].
\end{aligned}$$

$\square$

## D.3 CONVERGENCE GUARANTEE

We can now prove the convergence guarantee reported in Equation (4). The proof of our result is obtained combining the analysis from the sublinear convergence guarantee with the analysis for the inexact linear convergence guarantee given by (Xiao, 2022, Theorems 8 and 13). For two different time $t, t' \geq 0$, denote the expected Bregman divergence between the policy $\pi^t$ and policy $\pi^{t'}$, where the expectation is taken over the discounted state visitation distribution of the optimal policy $d_\mu^\star$, by

$$\mathcal{D}_{t'}^t := \mathbb{E}_{s \sim d_\mu^\star} \left[ \mathcal{D}_h(\pi_s^t, \pi_s^{t'}) \right].$$

Similarly, denote the expected Bregman divergence between the optimal policy $\pi^\star$ and $\pi^t$ by

$$\mathcal{D}_t^\star := \mathbb{E}_{s \sim d_\mu^\star} \left[ \mathcal{D}_h(\pi_s^\star, \pi_s^t) \right].$$

**Theorem D.4** (Theorems 8 and 13 in Xiao (2022))**.** *Consider the PMD update in (2), at each iteration $T \geq 0$, we have*

$$V^\star(\mu) - \sum_{t<T} \mathbb{E}[V^t(\mu)] \leq \frac{1}{T} \left( \frac{\mathbb{E}[\mathcal{D}_0^\star]}{\eta_t(1-\gamma)} + \frac{1}{(1-\gamma)^2} \right) + \frac{4}{(1-\gamma)^2} \mathbb{E} \left[ \max_{s \in \mathcal{S}} \left\| \widehat{Q}_s^t - Q_s^t \right\|_\infty \right].$$

*Proof.* Using Lemma D.2 with $x^+ = \pi_s^{t+1}$, $\mathcal{C} = \Delta(\mathcal{A})$, $f(u) = \langle \widehat{Q}_s^t, u \rangle$, $x = \pi^t$ and $u = \pi^{t+1}$, we have that

$$\langle \eta_t \widehat{Q}_s^t, \pi_s^\star - \pi_s^{t+1} \rangle \leq \mathcal{D}_h(\pi^\star, \pi^t) - \mathcal{D}_h(\pi^\star, \pi^{t+1}) - \mathcal{D}_h(\pi^{t+1}, \pi^t),$$

which can be decomposed as

$$\langle \eta_t \widehat{Q}_s^t, \pi_s^t - \pi_s^{t+1} \rangle + \langle \eta_t \widehat{Q}_s^t, \pi_s^\star - \pi_s^t \rangle \le \mathcal{D}_h(\pi^\star, \pi^t) - \mathcal{D}_h(\pi^\star, \pi^{t+1}) - \mathcal{D}_h(\pi^{t+1}, \pi^t).$$

Taking expectation with respect to the distribution $d_\mu^\star$ over states and with respect to the randomness of PMD and dividing both sides by $\eta_t$, we have

$$\mathbb{E}\left[\mathbb{E}_{s \sim d_\mu^\star}\left[\langle \widehat{Q}_s^t, \pi_s^t - \pi_s^{t+1} \rangle\right]\right] + \mathbb{E}\left[\mathbb{E}_{s \sim d_\mu^\star}\left[\langle \widehat{Q}_s^t, \pi_s^\star - \pi_s^t \rangle\right]\right] \le \frac{1}{\eta_t} \mathbb{E}[\mathcal{D}_t^\star - \mathcal{D}_{t+1}^\star - \mathcal{D}_t^{t+1}]. \quad (13)$$

We lower bound the two terms on the left hand side of (13) separately. For the first term, we have that

$$\mathbb{E}\left[\mathbb{E}_{s \sim d_\mu^\star}\left[\langle \widehat{Q}_s^t, \pi_s^t - \pi_s^{t+1} \rangle\right]\right] \overset{(a)}{\ge} \frac{1}{1-\gamma} \mathbb{E}\left[\mathbb{E}_{s \sim d_{d_\mu^\star}^{t+1}}\left[\langle \widehat{Q}_s^t, \pi_s^t - \pi_s^{t+1} \rangle\right]\right]$$

$$= \frac{1}{1-\gamma} \mathbb{E}\left[\mathbb{E}_{s \sim d_{d_\mu^\star}^{t+1}}\left[\langle Q_s^t, \pi_s^t - \pi_s^{t+1} \rangle\right]\right] + \frac{1}{1-\gamma} \mathbb{E}\left[\mathbb{E}_{s \sim d_{d_\mu^\star}^{t+1}}\left[\langle \widehat{Q}_s^t - Q_s^t, \pi_s^t - \pi_s^{t+1} \rangle\right]\right]$$

$$\overset{(b)}{=} \mathbb{E}\left[V^t(d_\mu^\star) - V^{t+1}(d_\mu^\star)\right] + \frac{1}{1-\gamma} \mathbb{E}\left[\mathbb{E}_{s \sim d_{d_\mu^\star}^{t+1}}\left[\langle \widehat{Q}_s^t - Q_s^t, \pi_s^t - \pi_s^{t+1} \rangle\right]\right]$$

$$\ge \mathbb{E}\left[V^t(d_\mu^\star) - V^{t+1}(d_\mu^\star)\right] - \frac{1}{1-\gamma} \mathbb{E}\left[\mathbb{E}_{s \sim d_{d_\mu^\star}^{t+1}}\left[\left\|\widehat{Q}_s^t - Q_s^t\right\|_\infty \left\|\pi_s^{t+1} - \pi_s^t\right\|_1\right]\right]$$

$$\ge \mathbb{E}\left[V^t(d_\mu^\star) - V^{t+1}(d_\mu^\star)\right] - \frac{2}{1-\gamma} \mathbb{E}\left[\max_{s \in \mathcal{S}}\left\|\widehat{Q}_s^t - Q_s^t\right\|_\infty\right]$$

where $(a)$ follows from Lemmas D.3 and the fact that $d_{d_\mu^\star}^{t+1}(s) \ge (1-\gamma) d_\mu^\star(s) \ \forall s \in \mathcal{S}$, and $(b)$ follows from D.1. For the second term, we have that

$$\mathbb{E}\left[\mathbb{E}_{s \sim d_\mu^\star}\left[\langle \widehat{Q}_s^t, \pi_s^\star - \pi_s^t \rangle\right]\right] = \mathbb{E}\left[\mathbb{E}_{s \sim d_\mu^\star}\left[\langle Q_s^t, \pi_s^\star - \pi_s^t \rangle\right]\right] + \mathbb{E}\left[\mathbb{E}_{s \sim d_\mu^\star}\left[\langle \widehat{Q}_s^t - Q_s^t, \pi_s^\star - \pi_s^t \rangle\right]\right]$$

$$\overset{(b)}{=} \mathbb{E}[V^\star(\mu) - V^t(\mu)](1-\gamma) + \mathbb{E}\left[\mathbb{E}_{s \sim d_\mu^\star}\left[\langle \widehat{Q}_s^t - Q_s^t, \pi_s^\star - \pi_s^t \rangle\right]\right],$$

$$\ge \mathbb{E}[V^\star(\mu) - V^t(\mu)](1-\gamma) - \frac{2}{1-\gamma} \mathbb{E}\left[\max_{s \in \mathcal{S}}\left\|\widehat{Q}_s^t - Q_s^t\right\|_\infty\right],$$

where $(b)$ follows from Lemma D.1.

Plugging the two bounds in (13), dividing both sides by $(1-\gamma)$ and rearranging, we obtain

$$\frac{\mathbb{E}[\mathcal{D}_t^{t+1}]}{\eta_t(1-\gamma)} + \mathbb{E}[V^\star(\mu) - V^t(\mu)] \le \frac{\mathbb{E}[\mathcal{D}_t^\star - \mathcal{D}_{t+1}^\star]}{\eta_t(1-\gamma)} + \frac{\mathbb{E}\left[V^t(d_\mu^\star) - V^{t+1}(d_\mu^\star)\right]}{1-\gamma}$$

$$+ \frac{4}{(1-\gamma)^2} \mathbb{E}\left[\max_{s \in \mathcal{S}}\left\|\widehat{Q}_s^t - Q_s^t\right\|_\infty\right].$$

Summing up from 0 to $T-1$ and dropping some positive terms on the left hand side and some negative terms on the right hand side, we have

$$TV^\star(\mu) - \sum_{t<T} \mathbb{E}[V^t(\mu)] \le \frac{\mathbb{E}[\mathcal{D}_0^\star]}{\eta_t(1-\gamma)} - \frac{\mathbb{E}\left[V^0(d_\mu^\star) - V^T(d_\mu^\star)\right]}{1-\gamma}$$

$$+ \frac{4}{(1-\gamma)^2} \mathbb{E}\left[\max_{s \in \mathcal{S}}\left\|\widehat{Q}_s^t - Q_s^t\right\|_\infty\right].$$

Notice that $\mathbb{E}\left[V^0(d_\mu^\star) - V^T(d_\mu^\star)\right] \le \frac{1}{1-\gamma}$ as $r(s,a) \in [0,1]$. By dividing $T$ on both side, we yield the statement. $\qquad \square$

# E TRAINING DETAILS

We give the hyper-parameters we use for training in Tables 2, 3 and 4. Hyper-parameter tuning was performed differently for each method. For Gridworld, we performed a grid search over the hyper-parameters and selected those that maximized the averaged performance of the negative entropy

and $\ell_2$-norm over 256 randomly sampled Gridworld environments. For Basic Control Suite and MinAtar, we used the optuna library to search over the hyper-parameters to maximize the average performance of the negative entropy over all environments. We used "CmaEsSampler" as option for the sampler. Lastly, we used Weights and Biases to optimize the hyperparameters for MuJoCo in order to maximimize the average performance of the negative entropy over all environments.

Table 2: Hyper-parameter settings of PMD for different sets of environments

| Parameter | Grid-World | MuJoCo |
|---|---|---|
| Number of environment steps | $2^{18} \simeq 2.5e5$ | 1e7 |
| Number of environments | 64 | 2048 |
| Unroll length | 32 | 10 |
| Number of minibatches | 1 | 128 |
| Number of update epochs | 32 | 8 |
| Adam learning rate | - | 1e-4 |
| Sgd learning rate | 40 | - |
| Gamma | 0.99 | 0.99 |
| Max grad norm | - | 1.0 |
| $\eta$ | 0.1 | 0.5 |

Table 3: Hyper-parameter settings of AMPO for different sets of environments

| Parameter | BCS | MinAtar |
|---|---|---|
| Number of environment steps | 5e5 | 1e7 |
| Number of environments | 4 | 256 |
| Unroll length | 128 | 128 |
| Number of minibatches | 4 | 8 |
| Number of update epochs | 16 | 8 |
| Adam learning rate | 4e-3 | 7e-4 |
| Gamma | 0.99 | 0.99 |
| Max grad norm | 1.4 | 1 |
| AMPO learning rate | 0.9 | 0.9 |

Table 4: Hyper-parameter settings of OpenAI-ES, which we used in the tabular setting, and of Sep-CMA-ES, which we used in the non-tabular case.

| | OpenAI-ES | Sep-CMA-ES |
|---|---|---|
| Population Size | 512 | 128 |
| Number of generations | 512 (256 for MuJoCo) | 600 |
| Sigma init | 0.5 | 2 |
| Sigma Decay | 0.995 | - |
| Learning rate | 0.01 | - |

# F  COMPUTATIONAL COSTS OF ES

The computational costs associated with ES are inevitable in this field of research that focuses on automatically discovering optimizers. In the literature, the most popular methods to discover optimizers are meta-gradients (Oh et al., 2020; Jackson et al., 2023) and ES (Lu et al., 2022). Discovering algorithms using meta-gradients consists in introducing some meta-parameters that influence the agent training procedure, training a batch of agents, and differentiating through the training procedure w.r.t. the meta-parameters to maximize the final performance of the agents. ES consists in running several training procedures in parallel and estimating the meta-gradient as in (9), therefore avoiding the need for differentiation. This property is particularly desirable in settings, like ours, where the agent is updated many times and differentiating through the whole training procedure is unfeasible due to memory constraints, meaning that it is necessary to limit the differentiation to the final updates. We decided to employ ES due to this property and because it has been found to be more successful in discovering optimizers (Jackson et al., 2024). Additionally, our experiments are implemented in JAX, which through parallelism and just in time compilation renders the computational costs of ES feasible.

