# OpenReview forum: "Learning mirror maps in policy mirror descent"
_ICLR.cc/2025/Conference — ICLR 2025 Poster_

### Official Review · Reviewer_CN9i · 2024-11-03

**Soundness:** 3
**Presentation:** 3
**Contribution:** 3
**Rating:** 8
**Confidence:** 4

**Summary:**

This paper investigates the Policy Mirror Descent in RL, which is known for its finite-time convergence guarantees and encompasses various algorithms by selecting a mirror map. However, the study explores the potential of alternative mirror maps (instead of negative entropy mirror map, which leads to the Natural Policy Gradient method), demonstrating that the conventional choice often results in suboptimal performance. In empirical studies, the authors use evolutionary strategies to identify more effective mirror maps, showing improved performance in tabular and non-tabular environments. The research highlights that the choice of mirror map significantly affects PMD's effectiveness, challenging existing theoretical assumptions and suggesting new directions for future research.

**Strengths:**

1. The research effectively highlights the limitations of existing convergence guarantees in capturing the impact of mirror maps. It provides robust empirical evidence that challenges the conventional choice of mirror maps, offering a fresh perspective on the potential of PMD.

2. The experimental design is well-conceived and rigorous, aligning closely with the preceding analysis.

3. The paper is well-written, with clear definitions and precise descriptions. Despite covering a substantial amount of theory, it remains accessible and easy to read.

**Weaknesses:**

From my perspective, this paper makes a satisfactory and engaging contribution, though there is always room for improvement.

1. The paper offers compelling empirical evidence but does not provide a comprehensive theoretical framework to elucidate why certain mirror maps outperform others. Although developing such a framework is challenging, it represents a promising avenue for future research.

2. the study's focus on specific benchmark environments may not fully encompass the diversity and complexity of real-world scenarios in reinforcement learning, potentially limiting the generalizability of the findings.

**Questions:**

Q1: Although the experiments demonstrate that the learned map is more effective, I believe this does not necessarily conflict with the theoretical assertion that these methods share the same bound. Could the authors provide further explanation on this point?

Q2: In lines 165-166, what is the relationship between Bregman divergence and the mirror map $h$? Could this be further clarified?

Q3(minor): There is a typo in Equation 5 (\theta).

---

> ### Author Response · Authors · 2024-11-23
> **Author reply**
>
> Thank you for your positive and encouraging remarks. Please find answers to your questions below.
>
> **[Convergence bounds]** We agree with the reviewer that our empirical results do not contradict the theoretical assertion that all PMD algorithms share the same upper bound in equation (4), especially because the assumptions behind Theorem 2.2 are satisfied by the tabular GridWorld environment. However, our experiments suggest that this bound is loose and fails to tightly capture the behavior of different mirror maps. Additionally, while the bound suggests that the estimation error influences the performance of PMD, we have found this to be false in practice (as shown in Figure 2 and discussed in Lines 376-377).
>
> **[Connection between mirror map and Bregman divergence]** We now provide more intuition behind the connection between the mirror map and the associated Bregman divergence in the Appendix.

---

> > ### Comment · Reviewer_CN9i · 2024-11-27
> > **Response to Authors**
> >
> > Thanks for the rebuttal. The authors have addressed my questions.

---

### Official Review · Reviewer_4HAM · 2024-11-03

**Soundness:** 2
**Presentation:** 4
**Contribution:** 3
**Rating:** 6
**Confidence:** 3

**Summary:**

This paper explores the role of mirror maps in the Policy Mirror Descent (PMD) framework in reinforcement learning (RL), which includes various algorithms for policy optimization. Traditionally, the negative entropy mirror map, giving rise to the Natural Policy Gradient (NPG), is the most studied, but the authors argue that this choice is often sub-optimal. Through empirical experiments, they show that alternative, learned mirror maps can enhance PMD's performance across different RL environments, starting with tabular settings like Grid-World and scaling to more complex scenarios like MinAtar and MuJoCo. Their findings suggest that mirror map choice has a more substantial impact on PMD performance than theoretical convergence guarantees indicate. The manuscript highlights the potential for learned mirror maps to generalize across tasks and calls for further theoretical research to understand the influence of mirror maps on PMD's effectiveness.

**Strengths:**

- Empirical validation: the authors provide empirical evidence that challenges the reliance on the negative entropy mirror map in PMD.
- Comparison with theory: the paper highlights a gap between theoretical bounds and actual empirical performance.
- Algorithm: the use of ES makes sense here and is well justified.
- Clarity: the background and methodology sections are well written.

**Weaknesses:**

- Reliance on ES which may not be very scalable (to higher dimensional mirror map parameterization).
- Experiments only include small environments where finding better mirror maps may not lead to significantly superior learning curves. Testing on a broader array of environments, especially those with different reward structures or high-dimensional state spaces would enhance the robustness and applicability of the findings.
- Experiments do not seem statistically significant and may hinder hyper-parameter optimization for some methods and not e.g. for the negative entropy mirror map baseline.

**Questions:**

1. How does the mirror map learned on e.g. Hopper perform on very different environments (other than Mujoco's)? Is performance/training dynamics significantly different from using the negative entropy mirror map?
2. I find it surprising that using negative entropy or l2 norm does not solve Asterix-MinAtar under 10M steps. I personally recall experimenting with Asterix and rather easily solving it. Could you provide a more detailed description of how hyperparameters were selected for each method, or include ablation studies showing the impact of key hyperparameters?

---

> ### Author Response · Authors · 2024-11-23
> **Author reply**
>
> Thank you for your review and remarks. Please find replies to your questions in the general comment and below.
>
> **[Environments]** The primary goal of our paper is to demonstrate that, contrary to the implications of current theoretical results based on upper bounds, different mirror maps can result in significantly varied performances of policy mirror descent (PMD) algorithms. In particular, our aim is to (1) provide an intuition on the cause of this behavior and (2) show that the negative entropy is often not the optimal choice of mirror map. To achieve the first objective, it is necessary to study simple tabular environments, as we need access to ground truth quantities, such as the Q-value function, to relate the behavior of PMD with its theoretical guarantees. As to the second objective, we have chosen our benchmark environments following the literature on automatic discovery of reinforcement learning algorithms [1,2], where the efficacy of the proposed methods is established through benchmarking on Gridworld, MinAtar or MuJoCo environments, as we do.
>
> **[Improved Performance and Statistical Significance]** We would like to highlight that the performance of our discovered mirror maps improves upon the negative entropy by at least 10\% in most environments, reaching 50\% in the "Labyrinth", "Small corridor" and "Standard maze", and 100\% in "Asterix" and "SpaceInvaders", as reported in Figures 1, 2, 4 and in Table 3. Following common practice [1,2], we report both the average performance and its standard error as a shaded region for all our plots to confirm the statistical significance of our results. We note that the shaded region associated with the learned mirror map does not overlap with the one associated to the negative entropy in almost all plots, pointing to the statistical significance of our results. Lastly, we note that the performance improvement over the negative entropy that we report for our discovered mirrored maps is similar or larger than the performance improvement reported by previous works on automatic discovery of machine learning algorithms [1,2,3], where the improvement over the respective baseline is usually around 10\% to 20\%.
>
> **[Inter-suite transfer]** We note that in the paper we already report the performance of mirror maps learned on Basic Control Suite and tested on MinAtar and viceversa in Table 1. While they are very different environments, we still see that the learned maps generalize well and outperform the negative entropy in most cases. Following the reviewer's suggestion, we have tested PMD with both the negative entropy and the mirror map learned for Hopper on SpaceInvaders, obtaining final rewards of 167.32 and 171.63 respectively, averaged over 20 runs. While the performance does not increase, the fact that is does not decrease even if the mirror map was optimized for a completely different environment (discrete vs continuous action spaces) testifies to the robustness capabilities of our proposed methodology. We would also like to remind the reviewer that our objective is not to show that there exists a mirror map that is always superior to the negative entropy, but that in most environments it is possible to find a mirror map that outperforms the negative entropy.
>
> **[Performance on Asterix]** We would like to note that we are using the MinAtar version of Asterix from the gymnax library, which has a different structure from the standard Asterix environment. The performance that we report for AMPO with the negative entropy is similar to the one reported by the library authors for PPO, that is around 15 after 10M steps.
>
> **[Hyper-parameter tuning]** Hyper-parameter tuning was performed differently for each method. For Gridworld, we performed a grid search over the hyper-parameters and selected those that maximized the averaged performance of the negative entropy and $\ell_2$-norm over 256 randomly sampled Gridworld environments. For Basic Control Suite and MinAtar, we used the optuna library to search over the hyper-parameters to maximize the average performance of the negative entropy over all environments. We used "CmaEsSampler" as option for the sampler. Lastly, we used Weights and Biases to optimize the hyperparameters for MuJoCo in order to maximimize the average performance of the negative entropy over all environments. We used "bayes" as option for the method. We now include this more in detailed discussion in the Appendix.
>
> [1] Lu, Chris, et al. "Discovered policy optimisation." Advances in Neural Information Processing Systems 35 (2022).
>
> [2] Jackson, Matthew Thomas, et al. "Discovering Temporally-Aware Reinforcement Learning Algorithms." The Twelfth International Conference on Learning Representations (2024).
>
> [3] Lu, Chris, et al. "Discovering Preference Optimization Algorithms with and for Large Language Models." Automated Reinforcement Learning: Exploring Meta-Learning, AutoML, and LLMs.

---

> > ### Comment · Reviewer_4HAM · 2024-11-24
> >
> > Thank you for providing complete answers to my questions and clarifying most points of concern.
> >
> > I believe the work brings a significant contribution to the field and after much consideration, I am changing my rating from 5 to 6.
> >
> > However, I still think the experiments in this work could be improved in many ways as mentioned above.

---

> > > ### Author Response · Authors · 2024-11-30
> > > **Author reply**
> > >
> > > We are grateful to the reviewer for taking the time to reassess our work and increasing the score. We have updated the paper to reflect the comments made in the rebuttal, including the discussion on the hyper-parameter tuning procedure and the discussion on the computational costs of ES.

---

### Official Review · Reviewer_WJaW · 2024-11-04

**Soundness:** 3
**Presentation:** 2
**Contribution:** 3
**Rating:** 6
**Confidence:** 3

**Summary:**

This paper studies the impact of different mirror graphs within the Policy Mirror Descent (PMD) framework in reinforcement learning, traditionally dominated by negentropy graphs. Through empirical testing, the authors identify alternative mirror mappings that significantly improve performance and generalization in different environments, challenging the default choice of mirror mappings in PMD.

**Strengths:**

1. The author presents a novel approach by replacing the conventional mirror map with a learned mirror map, offering valuable insights for reinforcement learning research.
2. The experiments cover both a tabular setting and a real-world application, enhancing the potential for this work to be applied to real-world problems.

**Weaknesses:**

1. There are some typos in this paper, for example, in equation (5), it should be $\pi_{\theta_t}$ instead of $\pi_{t}heta$?
2. The author didn't provide related works of this paper, making it hard to understand and follow. And it is better to provide more training details in Appendix C.
3. It seems that the author only provides comparisons of their method with the negative entropy and d $\ell_2$-norm mirror map. Can the author provide comparisons with other methods?
4. The computational cost of the proposed method appears to be relatively high. It would be beneficial for the author to include a discussion comparing the computational cost of this method to that of existing approaches. This comparison would provide a valuable disscussion for understanding the trade-offs in efficiency and practicality.

**Questions:**

See weaknesses.

---

> ### Author Response · Authors · 2024-11-23
> **Author reply**
>
> Thank you for your review and remarks. Please find replies to your questions in the general comment and below.
>
> **[Related work]** We have now included a related work section, discussing other works that focus on discovering reinforcement learning optimizers. We would also like to highlight that the main contribution of the paper is to the literature on Policy Mirror Descent, which is discussed in the introduction.
>
>
> **[Chosen baselines]** We chose the negative entropy and the $\ell_2$-norm as baselines mirror maps as they are representative of several algorithms in the literature. PMD with the negative entropy, which includes a soft KL-divergence penalty in the update rule, is a foundational algorithm representing the backbone of TPRO and PPO, whereby TRPO enforces a hard KL-divergence constraint and PPO approximates the soft KL-divergence penalty through a clipped objective. PMD with the negative entropy is also linked to Natural Policy Gradient, to which is equivalent in the tabular setting. We include the $\ell_2$-norm as a baseline due fact that Mirror Descent with the $\ell_2$-norm recovers the classical gradient descent method.

---

> > ### Comment · Reviewer_WJaW · 2024-11-26
> >
> > Thanks for your reply. It addresses some of my questions. I will adjust my score accordingly.

---

> > > ### Author Response · Authors · 2024-11-30
> > > **Author Reply**
> > >
> > > We thank the reviewer for positively reassessing our work. We have updated the paper to account for the comments made in the rebuttal, including the typo in equation (5), the discussion on the computational costs of ES, and the discussion on related work in the field of discovering reinforcement learning optimizers.

---

### Author Response · Authors · 2024-11-23
**General comment**

We would like to thank the reviewers for their careful remarks and for taking the time to review our paper. In particular, we thank Reviewers *CN9i* and *4HAM* for recognizing that our work challenges conventional choices in PMD and that our paper is well written. We also thank Reviewer *WJaW* for acknowledging that our work offers valuable insights. We address here a common concern on the computational costs of ES and reply to reviewer specific questions below.


**[Computational costs of ES]** The computational costs associated with ES are inevitable in this field of research that focuses on automatically discovering optimizers. In the literature, the most popular methods to discover optimizers are meta-gradients [1,2] and ES [3]. Discovering algorithms using meta-gradients consists in introducing some meta-parameters that influence the agent training procedure, training a batch of agents, and differentiating through the training procedure w.r.t. the meta-parameters to maximize the final performance of the agents. ES consists in estimating this gradient by running several training procedures in parallel and estimating the gradient as in Line 263 of our paper, therefore avoiding the need for differentiation. The two methods have a similar computational cost, as both require training multiple agents in parallel to update the meta-parameters. We decided to employ ES as it has been found to be more successful in discovering optimizers [4]. Additionally, our experiments are implemented in JAX, which through parallelism and just in time compilation renders the computational costs of ES feasible. As to the scalability of ES to higher dimensional mirror maps, we highlight that all maps belonging to the $\omega$-potential mirror map class can be efficiently parametrized by a single scalar function, i.e. the $\omega$-potential, regardless of the dimension of the action space.

[1] Oh, Junhyuk, et al. "Discovering reinforcement learning algorithms." Advances in Neural Information Processing Systems 33 (2020).

[2] Jackson, Matthew T., et al. "Discovering general reinforcement learning algorithms with adversarial environment design." Advances in Neural Information Processing Systems 36 (2023).

[3] Lu, Chris, et al. "Discovered policy optimisation." Advances in Neural Information Processing Systems 35 (2022).

[4] Jackson, Matthew Thomas, et al. "Discovering Temporally-Aware Reinforcement Learning Algorithms." The Twelfth International Conference on Learning Representations (2024).

---

### Meta-Review · Area_Chair_62Lb · 2024-12-23

**Metareview:**

In this paper, the authors argue that the well-studied negative entropy mirror map, which results in natural policy gradient, is often sub-optimal. They empirically show that learning a proper mirror map using extensive search methods, such as evolutionary strategies,
can result in improved performance in tabular and non-tabular environments. This work highlights the importance of the choice of mirror map in the performance of policy mirror descent algorithms and calls for further research to understand the influence of mirror maps on the effectiveness of these algorithms.

(+) The paper is well-written and easy to follow. The authors used clear definitions and precise descriptions, which helps their work to be accessible to a large audience.
(+) The experimental design is well-conceived and rigorous, and it is closely aligned closely with the analysis.
(+) It properly challenges the conventional choice of mirror maps and offers a fresh perspective on the potential of policy mirror descent algorithms.

Despite some issues (mainly minor) raised by the reviewers, which I strongly recommend that the authors address them, the paper could be useful for the community.

**Additional Comments On Reviewer Discussion:**

The authors successfully addressed most of the issues raised by the reviewers during the rebuttal.

---

### Decision · Program_Chairs · 2025-01-22

Accept (Poster)